# Secondary Structure of Subgenomic RNA M of SARS-CoV-2

**DOI:** 10.3390/v14020322

**Published:** 2022-02-04

**Authors:** Marta Soszynska-Jozwiak, Agnieszka Ruszkowska, Ryszard Kierzek, Collin A. O’Leary, Walter N. Moss, Elzbieta Kierzek

**Affiliations:** 1Institute of Bioorganic Chemistry, Polish Academy of Sciences, 61-704 Poznan, Poland; mjozwiak@ibch.poznan.pl (M.S.-J.); aruszkowska@ibch.poznan.pl (A.R.); rkierzek@ibch.poznan.pl (R.K.); 2Roy J. Carver Department of Biophysics, Biochemistry and Molecular Biology, Iowa State University, Ames, IA 50011, USA; caoleary@iastate.edu (C.A.O.); wmoss@iastate.edu (W.N.M.)

**Keywords:** RNA structure, SARS-CoV-2, COVID-19, subgenomic RNAs, sgRNA, sgRNA M, chemical mapping

## Abstract

SARS-CoV-2 belongs to the *Coronavirinae* family. Like other coronaviruses, SARS-CoV-2 is enveloped and possesses a positive-sense, single-stranded RNA genome of ~30 kb. Genomic RNA is used as the template for replication and transcription. During these processes, positive-sense genomic RNA (gRNA) and subgenomic RNAs (sgRNAs) are created. Several studies presented the importance of the genomic RNA secondary structure in SARS-CoV-2 replication. However, the structure of sgRNAs has remained largely unsolved so far. In this study, we probed the sgRNA M model of SARS-CoV-2 in vitro. The presented model molecule includes 5′UTR and a coding sequence of gene M. This is the first experimentally informed secondary structure model of sgRNA M, which presents features likely to be important in sgRNA M function. The knowledge of sgRNA M structure provides insights to better understand virus biology and could be used for designing new therapeutics.

## 1. Introduction

Severe acute respiratory syndrome coronavirus 2 (SARS-CoV-2) causes coronavirus disease 19 (COVID-19) and is responsible for widespread infection/death and concomitant disruptions to health services, travel, trade, education and has a negative impact on people’s physical and mental health [1]. SARS-CoV-2 belongs to the Betacoronavirus genus and is a member of the *Coronaviridae* family, which also includes alpha-, gamma- and deltacoronaviruses [2]. Like other coronaviruses, SARS-CoV-2 is enveloped and possess three structural proteins: membrane protein (M), spike protein (S), and envelope protein (E), while nucleocapsid protein (N) protects the viral RNA genome by forming a capsid [3]. SARS-CoV-2 also produces sixteen non-structural proteins (nsp1−16) and accessory proteins [4]. SARS-CoV-2 is an RNA virus and possesses a positive-sense, single-stranded RNA genome of ~30 kb [5]. The genomic RNA has a 5′ cap and 3′ polyA tail and is used for the production of two large overlapping polyproteins (pp): pp1a and pp1ab. Polyproteins pp1a and pp1ab contain the non-structural proteins 1–11 and 1–16, respectively. Many of them, with an N protein, create the replicase–transcriptase complex (RTC) [6]. Genomic RNA is used as the template for replication and transcription by RTC. These processes result in the generation of negative-sense RNA intermediates that serve as templates for the production of positive-sense genomic RNA (gRNA) and subgenomic RNAs (sgRNAs). The gRNA is packaged into progeny virions or is used for translation, while sgRNAs encode conserved structural proteins, nucleocapsid proteins and several accessory proteins [6,7,8,9].

In coronaviruses, each sgRNA possesses a short 5′-terminal leader sequence derived from the 5′ end of the genome. Transcription regulatory sequences (TRS) are necessary to add the leader sequence to sgRNAs [10]. TRSs are located at the 3′ end of the leader sequence (leader TRS, TRS-L), as well as upstream of the genes in the 3′-proximal part of the genome (body TRSs, TRS-B). TRSs contain a conserved 6–7 nt core sequence (CS) surrounded by variable sequences. During negative-strand synthesis, *RNA**-**dependent RNA polymerase* (RdRP) pauses when it crosses a body TRS and switches the template to the leader TRS. As a result, sgRNA possess a leader sequence derived from the 5′ untranslated region of the genome and a TRS 5′ of the open reading frame [11,12].

RNA secondary structure in untranslated and coding regions play a key role in the viral replication cycle [13]. The secondary structure within the RNA of SARS-CoV-2 has been intensively studied. Bioinformatic analysis showed that the SARS-CoV-2 genome has almost twice the propensity to form a secondary structure than one of the most structured RNA genomes in nature, the HCV genome [14,15]. Recently, detailed secondary structure models for the extended 5’UTR, frameshifting stimulation element, 3’UTR and regions of the SARS-CoV-2 viral genome that have a high propensity for RNA secondary structures and are conserved within SARS-CoV-2 strains were predicted [15,16] SARS-CoV-2 was also investigated for the presence of large-scale internal RNA base pairing (genome-scale ordered RNA structure) (GORS)) in its genome. This analysis showed the existence of 657 stem-loop structures and 2015 duplexes [17]. Data revealed that regions containing the highest amount of structure within the SARS-CoV-2 genome are in the 5′ end, as well as regions corresponding to glycoproteins S and M [18]. Recently, RNA structure probing of the full SARS-CoV-2 coronavirus genome both in vitro and in living infected cells was published [19,20,21,22,23]. Additionally, using RNA structure probing with nanopore direct-RNA sequencing, NAI reactivity values for 3a, E, M, 6, 7a, 7b, 8 and N sgRNAs were measured [23]. However, the structure model of sgRNA M has not yet been determined and discussed.

In this study, we performed probing of the sgRNA M sequence of SARS-CoV-2 in vitro. We used selective 2’-hydroxyl acylation analyzed by a primer extension (SHAPE) method and chemical mapping with dimethyl sulfate (DMS) and 1-cyclohexyl-(2-morpholinoethyl) carbodiimide metho-p-toluene sulfonate (CMCT) to obtain the secondary structure of sgRNA M. This is the first experimentally informed secondary structure model of sgRNA M, which presents features likely to be important in sgRNA M function. Several chemical reagents combined with a bioinformatics program allowed for the prediction of a high confidence structure. The comprehensive characterization of sgRNA M structure will provide important insights to better understand virus biology and could be used for designing new therapeutics.

## 2. Material and Method

### 2.1. Experimental Constructs

The DNA template for the synthesis of sgRNA M was obtained in several steps. Firstly, reverse transcription was carried out using SuperScript III (Thermo Fisher Scientific) with Random Primers Mix (New England BioLabs, Ipswich, MA, USA) on RNA of SARS-CoV-2 from strain Slovakia/SK-BMC5/2020 (received from https://www.european-virus-archive.com, accessed on: 9 July 2020). Next, three PCR reactions using cDNA as template and specific primers (F1 and RM, F2 and RM, F3 and RM, Table 1) were performed to amplify the TRS-M coding sequence and add the leader sequence and transcription promoter on the 5′ end of sgRNA M. After this step, primers FC and RC were used to add an EcoRI site on the 5′ end and a Pst I site on the 3′ end of the template of sgRNA M. DNA was purified using the Pure Link^TM^ PCR Micro Kit (Thermo Fisher Scientific, Waltham, MA, USA). The DNA template was cloned into pUC19 and sequenced using the M13F and M13R primers for confirmation of proper sequence (Table 1).

### 2.2. Oligonucleotides Synthesis and Labelling

Primers for reverse transcription were synthesized by the phosphoramidite approach on a MerMade synthesizer. Primers for reverse transcription were synthesized with fluorophores: 6-carboxyfluorescein (6-FAM) and 5-carboxy-4’,5’-dichloro-2’,7’-dimethoxyfluorescein (5-JOE) on the 5′-end (Table 2). Primers were deprotected and purified according to published protocols [24,25]. Concentrations of all oligonucleotides were measured using a Spectrophotometer UV (NanoDrop2000 Thermo Fisher Scientific, Waltham, MA, USA). Primers for PCR were purchased from Sigma-Aldrich, Saint Louis, MO, USA).

### 2.3. RNA Synthesis

The DNA template for in vitro transcriptions of sgRNA M was obtained by PCR from a modified puC19 plasmid using primers FM and RM (Table 1). DNA was purified using the Pure Link ^TM^ PCR Micro Kit (Thermo Fisher Scientific, Waltham, MA, USA). The in vitro transcription reaction was performed using a MEGAscript™ T7 Transcription Kit (Thermo Fisher Scientific) according to the manufacturer’s protocol. RNA product was purified using RNeasy MiniElute Cleanup Kit (Qiagen, Hilden, Germany). The integrity and purity of samples were checked on an agarose gel.

### 2.4. RNA Folding

Before each experiment, RNA was folded in the same manner. RNA was heated to 80 °C in water for 5 min and slowly cooled to 50 °C. At this temperature, folding buffer was added, and samples were slowly cooled to 37 °C. The final concentration of buffer was 300 mM NaCl, 5 mM MgCl_2_, 50 mM HEPES, pH 7.5. RNA integrity and homogeneity after folding were analyzed by native gel electrophoresis using 0.8% agarose gel running at 4  °C with low voltage. Under these conditions, one band was observed (Appendix A).

### 2.5. Chemical Mapping Using NMIA, DMS and CMCT

The folding of RNA was carried out as described above. Next, chemical mapping was conducted according to published procedures with appropriate optimizations [26,27,28]. Briefly, 5.6 mM of NMIA, 30 mM of CMCT or 0.18% of DMS were used in mapping reactions. Chemical mapping was performed at 37 °C with DMS, CMCT or NMIA for 15, 30 or 40 min, respectively. Parallel, control reactions were facilitated in the same condition but without mapping reagents. Modified nucleotides were read-out by primer extension using a stoichiometry of 2 pmol primer/2 pmol RNA. Primer extension was performed at 55 °C with reverse transcriptase SuperScript III (Thermo Fisher Scientific) using the manufacturer’s protocol. Next, cDNA fragments and ddNTP ladders were separated by capillary electrophoresis (Laboratory of Molecular Biology Techniques at Adam Mickiewicz University in Poznan). Primers labeled with 6-FAM were used for the detection of modification by DMS, CMCT or NMIA and with control reactions without mapping reagents. Samples were resolved in two capillaries (reaction and control) with ddNTP ladders. Primers labeled with 5-JOE were used for ddNTP ladders (most often ddATP). The experiments were performed in at least technical triplicate with the average results presented. To obtain the reactivity values of each nucleotide, the standard deviation (SD) was calculated (Appendix A).

### 2.6. Processing of Chemical Mapping Data

The QuShape program was used to analyze mapping data according to a published method [29]. NMIA reactivities were normalized by the QuShape program using model-free statistics to a scale spanning 0 to ∼2, where zero indicates no reactivity and 1.0 is the low average intensity for highly reactive RNA positions [29]. Nucleotides had normalized SHAPE reactivities 0–0.5, 0.5–0.7, and ≥0.7 correspond to unreactive, moderately reactive and highly reactive positions, respectively. Nucleotides with no data were designated as −999. Normalized SHAPE reactivities from the extension reaction of each primer were processed independently. DMS and CMCT modifications analysis was conducted similar to NMIA reactivity calculations, except that only strong modifications (reactivities ≥ 0.7) were used in RNAstructure program prediction.

Chemical mapping results were used in the RNAstructure program [30] for the prediction of the secondary structure of sgRNA M. Normalized SHAPE reactivity (as described above) were used in RNAstructure 6.2 through “Read SHAPE reactivity—pseudo free energy” mode with a slope of 1.8 and intercept of −0.6 kcal/mol [31]. DMS and CMCT strong reactivities were introduced in the same prediction using the “chemical modification” mode [32].

### 2.7. Bioinformatic Analysis of Base Pairs Probabilities

The sgRNA M base pair probabilities were obtained using the “Partition Function RNA” mode implemented in the RNAstructure program. SHAPE and chemical mapping experiment results were incorporated as constraints after loading the sequence file in “Partition Function RNA” mode, and a .pfs file was generated. All constraints were obtained as described in the *Processing of chemical mapping data* section and were the same as applied for sgRNA M folding. Next, the secondary structure of the sgRNA M model was annotated using the .pfs file using the “Add Probability Color Annotation” mode in the RNAstructure program, version 6.2

### 2.8. Covariation Analysis

The sequence of the in vitro probed sgRNA M underwent covariation analysis via the cm-builder pipeline, and details of this process are available [19]. Briefly, cm-builder utilizes the programs INFERence of RNA ALignment (INFERNAL) (here, release 1.1.2) [33,34] and R-scape (here, version 1.5.16) [35,36] to make alignments of a reference sequence to homologous sequences and then cross-evaluate a structural model for statistically significant covariation that maintains base pairing. The sgRNA M sequences were aligned to a previously generated fasta file [15] of 25,571 *Coronaviridae* sequences, obtained from the Virus Pathogen *Database* and Analysis Resource (ViPR database, https://www.viprbrc.org/brc/home.spg?decorator=vipr, accessed on 10 February 2021) [37,38].

Additionally, the sgRNA M sequence was queried against the nucleotide BLAST (Basic Local Alignment Search Tool, https://blast.ncbi.nlm.nih.gov/Blast.cgi, accessed on 10 December 2021) database and yielded 453 homologous sequences. These sequences were subsequently MAFFT (Multiple Alignment using Fast Fourier Transform) [39] aligned with the sgRNA M sequence. From here, alignments were used to calculate the conservation of nucleotides in each base pair of the model structure.

## 3. Results and Discussion

### 3.1. Structure Probing of sgRNA M of SARS-CoV-2

The sequence of sgRNA M of SARS-CoV-2 was obtained by adding the leader sequence to the M coding sequence from the SARS-CoV-2 strain Slovakia/SK-BMC5/2020 using PCR reactions. The sgRNA M sequence of our model is identical to the SARS-CoV-2 Wuhan-Hu-1 isolate (ID: NC_045512.2). A leader sequence is characteristic for the sgRNA of the Coranaviridae family, and “leader to body fusion” takes place during discontinuous transcription [12]. Chemical mapping was used to determine a secondary structure of sgRNA M. In vitro transcribed sgRNA M was folded in folding buffer (300 mM NaCl, 5 mM MgCl_2_, 50 mM 4-(2-hydroxyethyl)-1-piperazineethanesulfonic acid (HEPES), pH 7.5) to obtain a single RNA conformation, as assessed by non-denaturing agarose gel (Appendix A). Chemical mapping was performed at 37 °C with DMS (methylates N1 of A and N3 of C when unpaired), CMCT (modifies N3 of U and N1 of G when unpaired) and SHAPE reagent N-methylisatoic anhydride (NMIA) (modifies flexible 2′-hydroxyl groups on the ribose) [26,27,28]. The modifications from chemical mapping were analyzed by reverse transcription followed by capillary electrophoresis (Appendix A).

DMS strongly modified 123 nucleotides (nt), meaning that 32.6% of all adenosines and cytidines in sgRNA M were structurally accessible. CMCT modified 106 nts, which represents 40.9% of all uridines. The SHAPE reagent NMIA strongly modified 119 nts, and 56 nts had moderate modifications signals, representing 22.2% of all sgRNA M nucleotides (see Section 2.6). Chemical mapping data and SHAPE data are complementary to each other and together highlight the eight most flexible regions in sgRNA M (each containing at least seven flexible nucleotides in the sequence): 36–43, 76–86, 260–267, 469–481, 487–494, 596–603, 612–621, 729–743 (Appendix A).

### 3.2. Base Pair Probabilities

To assess prediction quality and identify well-defined structural regions, we calculated the secondary structure partition function using RNAstructure 6.2 and, from this, determined the base pair probabilities for model pairs [40]. For the partition function calculations, experimental data were included (see Section 2.7 for details). Results indicate that there are several regions with paired and unpaired nucleotides of more than 90% probability: 1–121, 131–210, 220–502, 707–766. Additionally, all single-stranded regions are well defined by having a low probability of pairing (Figure 1).

### 3.3. Model of Secondary Structure for sgRNA M

To predict the secondary structure of sgRNA M based on the experimental probing data, the results of chemical mapping were used to constrain predictions in the RNAstructure 6.2 program. SHAPE data were loaded as pseudoenergy constraints (the energy contribution of SHAPE reactive nt were penalized) and DMS and CMCT modifications were included as chemical mapping constraints (highly reactive nts are forbidden to be in Watson–Crick base pairs flanked by Watson–Crick base pairs). The default values for slope and intercept in the RNAstructure 6.2 program were used. The default values of these parameters were determined by optimizing the accurate modelling of the SHAPE data set with sequences of known structures [31].

Our model of sgRNA M is highly structured with plenty of accessible bulges and loops (Figure 2). RNA motifs in the sgRNA M model are thermodynamically stable and have high calculated base pair probability. The ΔG°_37_ of the entire folded secondary structure is −416 kcal/mol. We showed that most of the inaccessible regions defined by chemical mapping correspond to areas containing base pairs. The 5′end of the sgRNA M model was folded into three hairpins: SL1, SL2 and SL3. These three hairpins also occur in the 5′UTR of SARS-CoV-2 in its 5′ 300 nt fragment [20,41,42] and are present in in vitro models and in-cell models of the whole genome [20,21,22,23,43]. These hairpins are also in good agreement with a structural–phylogenetic analysis of group IIb coronaviruses [44] and in silico prediction of the whole SARS-CoV-2 genome [16,17]. Moreover, the folding of a 5′ leader sequence of sgRNA M is in agreement with a study of the secondary structure of sgRNA N [14]. This investigation showed that the 5′ leader sequence folds almost autonomously in the sgRNA N, with the exception of a few poorly determined long-range interactions [14]. SL1 is the most variable among SARS-CoV-2 variants [41], generally possessing mismatches, bulges and a high number of A–U and U–A base pairs. This fact causes less thermodynamic stability of SL1 than SL2 and SL3. On the other hand, this feature is important for the replication of mouse hepatitis virus (MHV), a well-studied member of the *Coronaviridae* family [45]. SL2 is conserved in all CoVs, typically containing a pentaloop stacked on a five base-pair stem and creating a U-turn motif. This hairpin plays a critical role in MHV replication and translation [46]. SL3 is conserved only in subgroups of beta and gammaCoVs [4] and contains TRS-L sequences that take part in discontinuous transcription [11,44].

Recently, a prediction of interaction between the SARS-CoV-2 genome and the human proteome indicated that a highly structured region at the 5′ end had a large number of interactions with proteins such as (1) ATP-dependent RNA helicase—DDX1, which was previously reported to be essential for Avian infectious bronchitis coronavirus replication [18,47], (2) adenosine deaminases acting on RNA (ADAR) that catalyzes the hydrolytic deamination of adenosine to inosine, which affects viral protein synthesis, proliferation and infectivity [18,48], and (3) 2′-5′-oligoadenylate synthetases which control viral RNA degradation [18,49,50]. Some of these proteins could interact with a leader sequence of sgRNA. This assumption was confirmed via experiments with DDX1 knockdown that reduced the number of sgRNA in SARS-CoV-1 infected cells [51]. This finding and the preservation of 5′UTR motifs in sgRNA M indicate similar interactions could occur with sgRNA M. Moreover, interactions between the SARS-CoV-2 genome, as well as sgRNAs and host RNAs, were revealed. However, the SARS-CoV-2 genome and sgRNAs take part in different interactions with host RNAs [23,43].

RNA structure probing coupled with nanopore direct-RNA sequencing were used to map sgRNAs with NAI in living cells, but the structure of sgRNA M was not proposed [23]. NMIA reactivities of sgRNA M mapped in vitro obtained herein were compared with the NAI reactivity of sgRNA M probed in infected Vero-E6 cells. Interestingly, the reactivity profile is similar for regions: 131–140, 160–172, 175–188, 219–228, 234–242, 245–252, 271–277, 279–291, 293–307, 342–357, 359–364, 366–374, 385–396, 401–406, 410–415, 419–426, 428–434, 438–445, 449–461, 494–499, 516–522, 525–530, 533–546, 559–566, 571–583, 585–601, 604–614, 630–637, 639–650, 680–692, 711–727, 733–743 (regions longer than five nucleotides were mentioned, nucleotides with reactivities 0–0.5, 0.5–0.7, and ≥0.7 were treated as unreactive, moderately reactive, and highly reactive positions, respectively). The reactivity of sgRNA M from living cells used in the RNAstructure program supported the existence of structural motifs in sgRNA M, such as 7–23, 131–150, 181–194, 231–252, 305–323, 340–363/434–458, 383–414, 517–590, 622–688, 712–758. The preservation of in vitro thermodynamically stable motifs in living cells could indicate that they are important in the viral replication cycle. On the other hand, some regions of sgRNA M from in vivo probing experiments have different reactivities than in vitro datasets, for example, 308–315, 322–332, 380–384. It is possible that these regions could interact with proteins or other cellular components.

We also compared long-range RNA–RNA interactions within the secondary structure of sgRNA M mapped in vitro (Figure 2) with the in vivo RNA–RNA interactome of sgRNA M [43]. Overall, these interactions are different and complex. Moreover, Ziv and coauthors discovered the co-existence of alternative SARS-CoV-2 gRNA and sgRNA topologies, held by long-range base-pairing between regions tens of thousands of nucleotides apart [43].

We additionally compared our sgRNA M model (Figure 2) with a proposed gene M secondary structure [20]. Generally, our presented model of the secondary structure of sgRNA M and the corresponding region of the whole SARS-CoV-2 genome obtained by probing in vivo [20] are different (Figure 3). This difference is in agreement with a previous study about the in vivo RNA–RNA interactome of the full-length SARS-CoV-2 genome and several sgRNAs. Here some structural aspects of viral RNA are also discussed [43]. This investigation revealed that the viral genome and subgenomes adopt alternative topologies inside the cell. Moreover, some long-range RNA–RNA interactions in sgRNA of SARS-CoV-2 are unique [43]. However, some regions in our presented model of sgRNA M fold in the same structure as in the genome M region of infected cells. Motifs: 153–167, 178–198, 223–259, 299–329, 340–350/448–458, 352–361/437–445, 365–433, 484–498, 514- 592, 604–698, 712–759 are consistent with the in-cell secondary structure model of the corresponding region in the SARS-CoV-2 genome [20]. However, hairpin 514–592 is three base pairs longer and possesses an additional internal loop in the cellular model. In turn, hairpins 131–150 and 281–297 are almost identical to those of the corresponding region in the SARS-CoV-2 genome that are mapped in cells [20]. In our model, loops are longer than the corresponding region of the in-cell model. On the other hand, the small motif 503–512 is the only hairpin structure uniquely characteristic for sgRNA M and does not exist in the context of the SARS-CoV-2 genome. Our determined sgRNA M secondary structure is also similar to corresponding regions of other published whole-genome SARS-CoV-2 models [22,23]. This similarity between our sgRNA M structure and the in-cell determined structure of the M sequence in the whole genome context is surprising since that RNA structure in vitro and in vivo could be significantly different. In vivo interaction between RNA and proteins or other molecules can influence secondary structure [52]. These data indicated that the sequence and thermodynamics alone are major determinants of sgRNA M structural motifs formation. All results point to these stable motifs being functionally significant. Furthermore, experiments and computational analyses have shown that large amounts of double-stranded regions have a strong propensity to interact with proteins and act as scaffolds for RNA-binding proteins ] [53,54,55]. sgRNA M is very structured, and it is possible for stable helices to interplay with proteins.

### 3.4. Local Structural Motifs in sgRNA M Are Mostly Independent of Leader Sequence

We compared our sgRNA M model with the corresponding region of an *in vitro* model of the whole SARS-CoV-2 genome [19] to check the influence of the 5′leader sequence on the folding of sgRNA. We indicated that the structures are different. This feature is consistent with a study of sgRNA N and its corresponding region in the SARS-CoV-2 genome. This data indicated that the same RNA sequences can fold in different structures in the subgenomic and genomic contexts [14]. However, some local motifs (Figure 2; 131–150, 153–167, 181–194, 223–259, 267–279/102–114, 280–297, 299–329, 331–335/101–105, 340–358/439–458, 372–426, 483–499, 514–592, 604–698, 712–758) within the sgRNA M and SARS-CoV-2 genome are identical. Motifs 604–698 is slightly different in both models. Similar motifs are stable independently of the neighboring regions (leader sequence in sgRNA M and sequences upstream/downstream of the M region in SARS-CoV-2 genome). It is possible that the existence and appropriate folding of neighboring regions has some, but relatively small, influence on some local motifs of sgRNA M. Some local structure motifs are identical within in vitro and in vivo SARS-CoV-2 genome models [20].

### 3.5. Covariation Analysis of the sgRNA M Secondary Structure

The covariation analysis of the sgRNA M model presented here, utilizing the cm-builder pipeline with an alignment against 25,571 *Coronaviridae* sequences, yielded no base pairs with statistically significant covariation. Covariation (i.e., sites of mutated sequence which maintain base pairing and a 2D structure) is often used to support the potential for an RNA motif to be functional as the structure is being preserved even when the primary sequence is not. Importantly though, a lack of covariation does not indicate a lack of potential functionality.

An analysis of the conservation of the sgRNA M model base pairs against an alignment of 453 homologous sequences showed an average base pair conservation of 97.57%, with stem SL1 (Figure 2) having an average conservation of 50.97%, stem SL2 (Figure 2) averaging 83.48% and the remaining base pairs averaging roughly 100% conservation (Appendix A).

The high degree of conservation of most base pairs in the model may be partially responsible for the lack of detectable covariation. This lack of significant covariation is in line with previous studies [15,19,56]. Despite extensive evidence of stable, ordered secondary structure, few motifs were supported by significant covariation. Additionally, as most of the structures presented here exist within the sgRNA M coding sequence, there are additional evolutionary pressures to conserve these sequences as to not disrupt the protein amino acid sequence.

### 3.6. Possible Influence of Nucleotide Mutation on sgRNA M Structure of SARS-CoV-2 Variants

The SARS-CoV-2 genome constantly evolves, new mutations appear and virus variants are monitored. Changing of RNA sequence must occur in a frame to not be lethal for the virus and to preserve the function of proteins and also RNA structure. Emerging mutations should retain base pairs in RNA structure motifs that are important in the viral cycle. Therefore, we analyzed nucleotide mutations in the sequence of the M gene of SARS-CoV-2 variants. Table 3 display mutations based on SARS-CoV-2 variants and their representatives deposited in Global Initiative on Sharing Avian Influenza Data (GISAID). When compared to our model, most of the nucleotide mutations occur in single-stranded regions of the sgRNA M structure and have no influence on the base-pairing structure. Interestingly, three consistent mutations (preserving base pairs) are in the helix region of motif 94–130/211–336. Consistent mutations are additional support for the existence of these long-distance interactions. Two hairpin motifs in regions 153–167 and 299–329 (Figure 2) appear to have no functional significance because non-consistent mutations in these structures exclude their conservation (Table 3).

## 4. Conclusions

For the first time, the secondary structure of sgRNA M was determined based on the experimental data from several chemical mapping methods and bioinformatic analyses. The secondary structure model contains unique features likely to be important for sgRNA M functions. The structure also includes several of the same motifs as the genomic M fragment in the SARS-CoV-2 genome (Figure 3). Previously published reactivity of sgRNA M from structure probing in living cells supported the existence of some of the presented structural motifs of our sgRNA M model [23]. Although covariation analysis shows no base pairs with statistically significant covariation, the mutations of gene M within SARS-CoV-2 variants is largely in agreement with the presented structure and supports long-distance helixes. This new knowledge about sgRNA M provides insights to better understand virus biology and could be used for anti-SARS-CoV-2 strategies and designing new therapeutics. The revealed unique or same as in gRNA M structural motifs could be promising targets for antisense oligonucleotides, siRNAs and small molecules

## Figures and Tables

**Figure 1 viruses-14-00322-f001:**
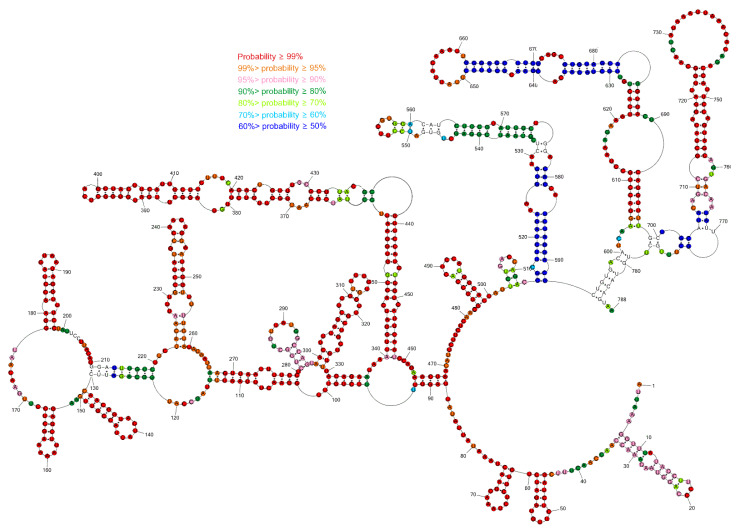
Predicted probability of nucleotides being paired or single-stranded in sgRNA M using the RNAstructure program. Probability lower than 50% is not colored. The partition function calculation incorporated restraints from strong reactivity of DMS and CMCT as well as SHAPE reactivities converted to pseudo-energies.

**Figure 2 viruses-14-00322-f002:**
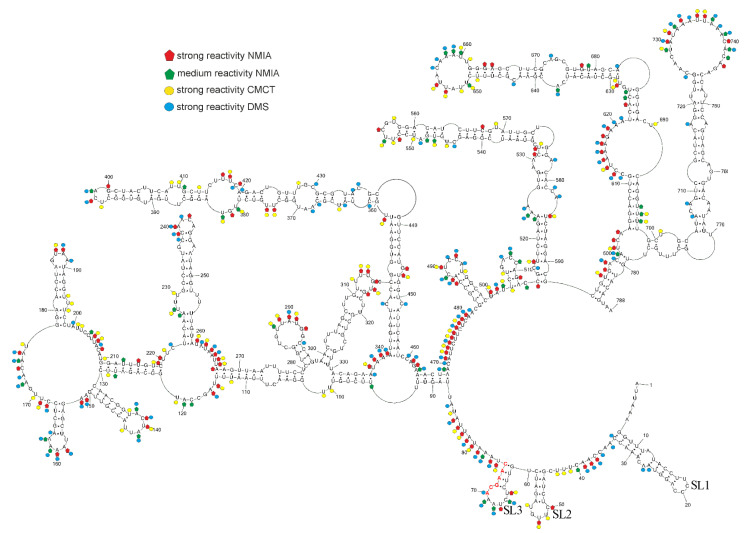
sgRNA M model predicted by RNAstructure 6.2 using experimental data as constraints. Strong DMS and CMCT modifications, as well as SHAPE reactivities converted to pseudo-free energies, were used. The numbering of sgRNA M is from its 5′ end. The AUG start codon spans nucleotides 120–122. Red nucleotides indicate TRS sequences. Hairpins SL1, SL2 and SL3, are indicated.

**Figure 3 viruses-14-00322-f003:**
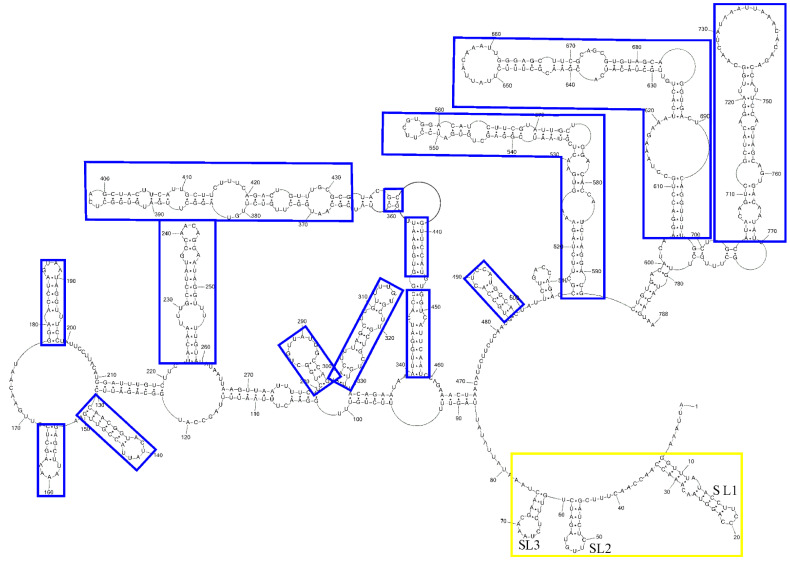
Comparison of the secondary structure of sgRNA M and its corresponding region of SARS-CoV-2 genome mapped in cells. Blue rectangle indicates the same base pairs within the sgRNA M model and the corresponding region of the SARS-CoV-2 genome mapped in cells [20]. Yellow rectangle indicates motifs of the leader sequence.

**Table 1 viruses-14-00322-t001:** Primers for polymerase chain reaction used to obtain DNA template for sgRNA M. The underlined nucleotide residues are the polymerase T7 promoter.

Primer Name	Primer Length (nt)	Sequence 5′→3′
FM	46	GCGTAATACGACTCACTATAGGGATTAAAGGTTTATACCTTCCCAG
RM	29	TTACTGTACAAGCAAAGCAATATTGTCAC
F1	54	CTTGTAGATCTGTTCTCTAAACGAACTAAATATTATATTAGTTTTTCTGTTTGG
F2	48	GTAACAAACCAACCAACTTTCGATCT CTTGTAGATCTGTTCTCTAAAC
F3	45	ATTAAAGGTTTATACCTTCCCAGGTAACAAACCAACCAACTTTCG
FC	29	TTCTGCAG ATTAAAGGTTTATACCTTCCC
RC	37	ATGAATTCTTACTGTACAAGCAAAGCAATATTGTCAC
M13F	24	CGCCAGGGTTTTCCCAGTCACGAC

**Table 2 viruses-14-00322-t002:** Primers for reverse transcription. Each primer was labeled with 6-FAM or 5-JOE at the 5′ end.

Primer Name	Primer Length (nt)	Sequence 5′→3′
M1	27	TTACTGTACAAGCAAAGCAATATTGTC
M2	21	CAGCTCCGATTACGAGTTCAC
M3	24	CAAGCTAAAGTTACTGGCCATAAC

**Table 3 viruses-14-00322-t003:** Mutations in the sgRNA M coding region in SARS-CoV-2 variants.

Nucleotide Mutations in Coding Region ^1^	SARS-CoV-2 Variants and Variants Representatives ^2^	Amino Acid Mutation ^3^	Influence of Nucleotide Mutations on RNA Structure ^4^
8(127) A > G	Omicron-B.1.1.529 (EPI ISL 6704867) Omicron BA.1 (EPI ISL 8482282) Omicron BA.2 (EPI ISL 8479001)	3D > G	consistent mutation A-U > G●U
46(165) C > A	AT.1 (EPI ISL 1652580)	16L > I	inconsistent mutationG-C > G..A
55(174) C > G	Omicron-B.1.1.529 (EPI ISL 6704867) Omicron BA.1 (EPI ISL 8482282) Omicron BA.2 (EPI ISL 8479001)	19Q > E	single-stranded region—no influence
82(201) U > C	R.1 (EPI ISL 1123466)	28F > L	single-stranded region—no influence
84(203) C > U	B.1.619.1 (EPI ISL 2361101)	-	single-stranded region—no influence
85(204) C > U	C.1.2 (EPI ISL 2942287)	29L > F	single-stranded region—no influence
159(278) C > U	A.2.5.2 (EPI ISL 1502915) Kappa-B.1.617.1 (EPI ISL 1384866) Epsilon-B.1.429 (EPI ISL 648527) Epsilon-B.1.427 (EPI ISL 1531902)	-	consistent mutation C-G > U●G
187(306) G > A	Omicron-B.1.1.529 (EPI ISL 6704867) Omicron BA.1 (EPI ISL 8482282) Omicron BA.2 (EPI ISL 8479001) AV.1 B.1.1.482.1 (EPI ISL 2179526)	63A > T	inconsistent mutations G-C > A..C
213(332) C > U	B.1.466.2 (EPI ISL 1533080)	-	consistent mutation C-G > U●G
241(361) G > U	Delta-AY.27 B.1.617.2 (EPI ISL 3910943)	81A > S	single-stranded region—no influence
245(364) U > C	Delta-B.1.617.2 (EPI ISL 1758376) C.1.2 (EPI ISL 2942287) B.1.640 (EPI ISL 5655471) Eta-B.1.525 (EPI ISL 760883) C.36.3 (EPI ISL 1245879) B.1.1.318 (EPI ISL 986813) B.1.619.1 (EPI ISL 2361101) B.1.575 (EPI ISL 2634469) B.1.1.523 (EPI ISL 2448704) AZ.5 (EPI ISL 2834919) Delta-B.1.617.2 (EPI ISL 2519798) Delta-AY.4.2 B.1.617.2 (EPI ISL 2851674) Delta-AY.33 B.1.617.2 (EPI ISL 4506526) Delta-AY.27 B.1.617.2 (EPI ISL 3910943) Delta-AY.26 B.1.617.2 (EPI ISL 2306061) Delta-AY.25 (EPI ISL 2295285) Delta-AY.3 B.1.617.2 (EPI ISL 3459265) Delta-AY.43 (EPI ISL 3565601) Delta-AY.47 (EPI ISL 2611181) Delta-AY.98.1 (EPI ISL 3305991) Delta-B.1.617.2 (EPI ISL 2306894)	82I > T	single-stranded region—no influence
245(364) U > G	Kappa-B.1.617.1 (EPI ISL 1384866)	82I > S	single-stranded region—no influence
279(398) C > G	B.1.177.82 (EPI ISL 617709)	-	single-stranded region—no influence
372(491) C > U	Lambda-C.37 (EPI ISL 1138413) N10 (EPI ISL 1465243)	-	single-stranded region—no influence
373(492) C > U	AV.1 B.1.1.482.1 (EPI ISL 2179526)	125H > Y	single-stranded region—no influence
450(569) U > C	B.1.258.17 (EPI ISL 618584)	-	single-stranded region—no influence
621(740) C > U	A.2.5.2 (EPI ISL 1502915)	-	single-stranded region—no influence

^1^—The first number in the column indicates the nucleotide number from the 5′ end of the M coding region; number in parenthesis means the corresponding nucleotide number of sgRNA M, as in Figure 2. ^2^—Variants and accession number of variant representatives in GISAID. ^3^—Number before mutation means amino acid number in M protein. ^4^—Consistent mutation means single point mutation that preserves pairing. “-“ indicates Watson-Crick pair, “●” shows GU wobble pair, “..” indicates no Watson-Crick and GU interactions.

## Data Availability

The data presented in this study are available in the article and Appendix A.

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
