# Peer review of "Secondary Structure of Subgenomic RNA M of SARS-CoV-2"

_viruses, 2022, doi:10.3390/v14020322_

Round 1

Reviewer 1 Report

All coronaviruses, including the pandemic strain SARS-CoV-2, are encoded by a translationally competent positive strand RNA genome. While the SARS-CoV-2 genome encodes for proteins directly, its negative strand also serves as the template for the generation of ten subgenomic mRNAs. These subgenomic RNAs each consist of an identical 5’ leader sequence, “stitched” to a subgenomic coding sequence. While the full genome, leader sequence, and sgRNAs have been structurally characterized in the past, very few reports focus on specific sgRNAs or their secondary structure modeling.

In this study, the authors construct one of these subgenomic RNAs (M; encoding for the membrane protein) using PCR and utilize three different RNA structure probing molecules (SHAPE, DMS, and CMCT) to infer its secondary structure in vitro. The use of three different structure probing molecules adds strength to the predicted model. However, the quality of the manuscript would be increased by attention to the comments below.

  1. The article is very brief, and the results are reported in a straightforward and concise manner, however, the article would benefit from thorough copyediting throughout.

  1. There are several published RNA probing experiments that could be (and perhaps have been) used to infer a secondary structure model for the M sgRNA. The authors should at least compare the reactivities from Wan et al., (and/or any other available sgRNA probing data) to their own. Many of these other datasets were determined in vivo or in virions, so would be important points of comparison against the authors in vitro dataset.

  1. Section 2.1 – The authors should describe here, or in the methods, what criteria are being used to define a strong modification.

  1. Section 2.2 – In this section the partition function is used to predict pairing probability with experimental data included. It would be good to explain this process in more detail in the methods (were data implemented during partition function calculations the same way as during MFE predictions?). It would also be good to know what the probabilities were without experimental data, since the experimental data is designed to alter these probabilities.
    •  
  2. In figure 3, it appears that the only difference between the authors’ model and the corresponding structures in genomic context (blue boxes), are several longer-range base pairs intervening previously predicted structures. Are there support for these intervening base pairs in the SPLASH data from Wan et al or the COMRADES data from Ziv et al.? Are these base pairs conserved?

  1. Can the authors discuss in more detail which strain/sequence of the M coding sequence was used and why? Were any important variants considered? Why wasn’t any downstream sequence used in construction of the sgRNA? It is known that biologically, sgRNAs do not end at the stop codon (as suggested by the models in Figures 1-3) but contain all downstream sequence through the genome’s 3’UTR.

  1. Why were the chosen slope (1.8) and intercept (-0.6) used during pseudo free energy calculations? Were any other parameters tested?

  1. The authors suggest that many of the structures in their models are important for function. Is there any evidence that any of them have function? Have the authors tested their structural model (or individual structures) for secondary structure conservation using programs such as Infernal and R-Scape?

  1. Several references need to be updated to reflect that they are no longer bioRXivs but have been published.

Author Response

Dear Reviewer #1,

            I am sending the revised manuscript entitled "Secondary structure of subgenomic RNA M of SARS-CoV-2” (manuscript ID: viruses-1525823). All authors of manuscript find the comments of Reviewers very helpful and interesting.

            In manuscript revision greatly contributed three additional persons, who are now included as co-authors of manuscript: Agnieszka Ruszkowska, Walter N. Moss and Collin A. O’Leary.

Below you will find our comments to suggestions:

Reviewer #1

  1. The article is very brief, and the results are reported in a straightforward and concise manner, however, the article would benefit from thorough copyediting throughout.

We thank the Reviewer for a positive assessment and a useful comment. We revised manuscript and make all efforts to eliminate any grammar and punctuations errors and make sentences clearly understandable in whole manuscript. The text was revised to smooth English and to be well understood by readers. We would like to emphasis that two authors of manuscript who greatly contributed in revision of the manuscript are native English speakers. We also used professional English grammar checking program. We believe that revised manuscript is clearly written in fluent English.

  1. There are several published RNA probing experiments that could be (and perhaps have been) used to infer a secondary structure model for the M sgRNA. The authors should at least compare the reactivities from Wan et al., (and/or any other available sgRNA probing data) to their own. Many of these other datasets were determined in vivo or in virions, so would be important points of comparison against the authors in vitro dataset.

We thank the Reviewer for this valuable comment. The available datasets for sgRNAs are limited. We analyzed reactivities of sgRNA M from NAI mapping in leaving cells that was collected and published in paper Yang, S.L. et al. Nature Communication 12, 5113 (2021) (Wan et.al. paper) and the conclusion from the analysis was placed in the manuscript. Also, we discussed our structure with in vivo RNA-RNA interactome of sgRNA M (O. Ziv et al., Mol. Cell, 80, 6, 1067-1077.e5, (2020)). Agnieszka Ruszkowska contributed greatly in this analysis.

We added the following paragraphs in section 2.3:

“RNA structure probing coupled with nanopore direct-RNA sequencing were used to map sgRNAs with NAI in living cells, but the structure of sgRNA M was not pro-posed [24]. NMIA reactivities of sgRNA M mapped in vitro obtained herein were compared with NAI reactivity of sgRNA M probed in infected Vero-E6 cells. Interestingly, the reactivity profile is similar for regions: 131- 140, 160-172, 175-188, 219-228, 234-242, 245- 252, 271-277, 279-291, 293-307, 342-357, 359- 364, 366-374, 385-396, 401-406, 410-415, 419-426, 428-434, 438-445, 449-461, 494-499, 516-522, 525-530, 533-546, 559-566, 571-583, 585-601, 604-614, 630-637, 639-650, 680-692, 711-727, 733-743 (regions longer than 5 nucleotide were mentioned, nucleotides with reactivities 0–0.5, 0.5–0.7, and ≥0.7 were treated like unreactive, moderately reactive, and highly reactive positions, respectively). Reactivity of sgRNA M from living cells used in the RNAstructure program supported existence of structural motifs in sgRNA M, such as 7-23, 131-150, 181-194, 231-252, 305-323, 340-363/434-458, 383-414, 517-590, 622-688, 712-758 . Preservation of in vitro thermodynamically stable motif in living cells could indicated that they are important in viral replication cycle. On the other hand, some regions of sgRNA M from in vivo probing experiments have different reactivities than in vitro datasets, for example 308-315, 322-332, 380-384. It is possible that these regions could interact with proteins or other cellular components.

We also compared long-range RNA-RNA interactions within secondary structure of sgRNA M mapped in vitro (Figure 1) with in vivo RNA-RNA interactome of sgRNA M [32]. Overall, these interaction are different and complex. Moreover, Ziv and coauthors discovered the co-existence of alternative SARS-CoV-2 gRNA and sgRNA topologies, held by long-range base-pairing between regions tens of thousands of nucleotides apart [32].”

  1. Section 2.1 – The authors should describe here, or in the methods, what criteria are being used to define a strong modification.

We thank the reviewer for this comment. We changed description about this issue in Methods, section 4.6, and defined in clear way a strong modification. Also, we refer to section 4.6 in the paragraph of section 2.1.

  1. Section 2.2 – In this section the partition function is used to predict pairing probability with experimental data included. It would be good to explain this process in more detail in the methods (were data implemented during partition function calculations the same way as during MFE predictions?). It would also be good to know what the probabilities were without experimental data, since the experimental data is designed to alter these probabilities.

We thank the Reviewer for this advice that improved manuscript. We added separate point in Method section: 4.7. Bioinformatic analysis of base pairs probabilities. Also, in Result section 2.2 we refer to section 4.7 to allow readers easy finding details.

The additional section:

“4.7. Bioinformatic analysis of base pairs probabilities

The sgRNA M base pair probabilities were obtained using “Partition Function RNA” mode implemented in the RNAstructure program. SHAPE and chemical mapping experiment results were incorporated as constraints after loading sequence file in “Partition Function RNA” mode and .pfs file was generated. All constraints were obtained as described in Processing of chemical mapping data section and were the same as applied for sgRNA M folding. Next, the secondary structure of sgRNA M model was annotated using the .pfs file using mode “Add Probability Color Annotation” in the RNAstructure program.”

We also calculated base pairs probability of our sgRNA M structure model without implementation of constraints. The figure with annotated probability is in the manuscript submission files, as separate pdf for Reviewer only (Figure_for_Reviewer_only.pdf). The probability of the base pairing is lower than calculated with implementation of all experimental data. It is of course reasonable. Introduced experimental data from structural mapping (in our case from SHAPE method and chemical mapping with CMCT and DMS) reduced predicted structures by RNAstructure to that which are fully in agreement with the experimental data. It is known that experimental data greatly facilitate RNAstructure prediction and without such data, especially for long RNA, the program can predict wrong structure (D. H. Mathews, et. al., Proceedings of the National Academy of Sciences USA, 101, 7287-72922 (2004); K. E. Deigan, et. al., Proceedings of the National Academy of Sciences USA, 106, 97-102 (2009)). Partition function for calculating base pair probabilities is used to determine confidence in base pairs predicted by free energy minimization (D.H. Mathews. RNA, 10,1178-1190 (2004)). Calculation of the base pairs probability with implemented experimental constraints show the probability of base pairs in the set of structures that are in agreement with experimental data. This is a very useful information and such information we presented in the manuscript. We think that showing base pairs probability without implemented constraints (as in file Figure_for_Reviewer_only.pdf) is interesting, but is not such useful as with constraints (as we presented in Figure 2). Thanks to the Figure 2 we know how confident the predicted structure is, and each of its structural motifs.

  1. In figure 3, it appears that the only difference between the authors’ model and the corresponding structures in genomic context (blue boxes), are several longer-range base pairs intervening previously predicted structures. Are there support for these intervening base pairs in the SPLASH data from Wan et al or the COMRADES data from Ziv et al.? Are these base pairs conserved?

We thank the Reviewer for this comments. We analyzed datasets included in Ziv et. al. paper. Below is a paragraph that was added in manuscript:

“We also compared long-range RNA-RNA interactions within secondary structure of sgRNA M mapped in vitro (Figure 1) with in vivo RNA-RNA interactome of sgRNA M [32]. Overall, these interaction are different and complex. Moreover, Ziv and coauthors discovered the co-existence of alternative SARS-CoV-2 gRNA and sgRNA topologies, held by long-range base-pairing between regions tens of thousands of nucleotides apart [32].”

  1. Can the authors discuss in more detail which strain/sequence of the M coding sequence was used and why? Were any important variants considered? Why wasn’t any downstream sequence used in construction of the sgRNA? It is known that biologically, sgRNAs do not end at the stop codon (as suggested by the models in Figures 1-3) but contain all downstream sequence through the genome’s 3’UTR.

We thank the Reviewer of these comments that bring our attention to add more details in the manuscripts. The sequence of the sgRNA M model is identical to SARS-CoV-2 isolate Wuhan-Hu-1 (ID: NC_045512.2). We added this information in the manuscript and also more information about our model RNA. We choose this strain to study the reference SARS-CoV-2 sequence of sgRNA M. The strain that we have was one of the first isolates of SARS-CoV-2 that was NGS fully sequenced and available for us through https://www.european-virus-archive.com.

The full length sgRNA M counts 3500 nt. Refolding and structure prediction of this long RNA molecules in vitro provides difficulties and limits structural data reliability. Thus, in this work we have focused on 5’UTR and coding region of M gene to investigate if 5’UTR leader sequence, added by “leader to body fusion” may affect structure of gene M. Our model of sgRNA M allow to propose for the first time its secondary structure and discuss it with available in literature data. The secondary structure model contains unique features likely to be important for sgRNA M functions. The structure also includes several of the same motifs as the genomic M fragment in the SARS-CoV-2 genome. Also, published reactivity data of sgRNA M from NAI mapping in cellulo supported existence of some presented structural motifs of our sgRNA M model.

According to Reviewers advice we also conducted two large analysis and added two sections “Covariation analysis of the sgRNA M secondary structure” and “Possible influence of nucleotide mutation on ssgRNA M structure of SARS-CoV-2 variants” in the manuscript. Both analysis strength our prediction of secondary structure of sgRNA M.

Covariation analysis shows no base pairs with statistically significant covariation, but the high degree of conservation of most base pairs in the model may be partially responsible for the lack of detectable covariation. A lack of covariation therefore does not exclude potential functionality. Whereas SARS-CoV-2 variants mutations mostly are in agreement with predicted sgRNA M structure. This new knowledge about sgRNA M provides insights to better understand virus biology and could be used for anti-SARS-CoV-2 strategies and designing of new therapeutics. Knowledge about secondary structure of RNA is a first step of rational and successful design of inhibitors targeting RNA.

More details about new paragraphs is in the answer to comment #8.

Please also see the answer to comment #1 of Reviewer # 3:

Reviewer #3

  1. I wonder how impactful this will be on the field. The secondary structure of RNA of SARS-CoV-2 has been extensively studied. How does the secondary structure of sgRNA M advance the field? Has other sgRNA been used as a therapeutic target?

SARS-CoV-2 viral cycle fully depends on RNA. Successful disruption of any sgRNA or gRNA will result with disruption of certain step of viral replication and in consequences degreasing amount of new virus particles. sgRNA as separate, much shorter than whole SARS-CoV-2 genome RNA, is probably more available target for different strategies as antisense strategies using oligonucleotides and small molecules targeting RNA. In our research of different viral RNA, influenza virus RNA, we showed that RNA secondary structure has a great impact on effectiveness of antisense oligonucleotides, siRNA and triplex forming PNA (Michalak, P. et al. Sci. Rep. 9, 3801 (2019); Piasecka, J. et al. Mol. Ther. - Nucleic Acids 19, 627–642 (2020); Lenartowicz, E. et al. Nucleic Acid Ther. 26, 277–285 (2016); Szabat, M. et al. Pathogens 9, 1–19 (2020), Kesy, J. et al. Bioconjug. Chem. 30, 931–943 (2019), Soszynska-Jozwiak, M. et al. Sci. Rep. 7, 15041 (2017)). Therefore it is important to know the structure of sgRNA and influence of 5’UTR on structure of coding region. Our research presented in manuscript show specific structural futures that may be connected with sgRNA function. Knowledge about secondary structure of SARS-CoV-2 sgRNAs is important for understanding virus biology.

Our research also show that in vitro study of model sgRNA combining with bioinformatics analysis, covariation and variants mutations study could contribute largely to determination of key structural features of the RNA. It is a relatively simple and fast way to know better RNA biology and for designing new therapeutics. sgRNAs were not use so far as a target for possible therapeutics although now it is obvious that inhibitors that are designed to target structural motifs that are the same in gRNA will target also sgRNA. Such knowledge about similarities and specificity of certain motifs of sgRNA comparing to gRNA – that we provided in the manuscript- is important and useful in the field.

  1. Why were the chosen slope (1.8) and intercept (-0.6) used during pseudo free energy calculations? Were any other parameters tested?

We would like to explain that we used the default slope and intercept for SHAPE pseudo free energies in RNAstructure6.2. We added this information in manuscript text. These values were determined by a search over reasonable values in previous work (PNAS 110: 5498 (2013)), and were trained with 16 ncRNAs with known structure. These values of slope and intersept worked well for a test set of 6 ncRNAs with known structure, and are the values recommended. We did not change the default parameters of RNAstructure 6.2. The default numbers are usually the best choice as they are determined by authors of program based on model experiments.

  1. The authors suggest that many of the structures in their models are important for function. Is there any evidence that any of them have function? Have the authors tested their structural model (or individual structures) for secondary structure conservation using programs such as Infernal and R-Scape?

We thank the Reviewer for this valuable comments that in consequence improved greatly the manuscript. We conducted additional analysis:

1/ covariation analysis in collaboration with Walter N. Moss and Collin A. O’Leary from Iowa State University (USA)

2/ mutation analysis of M coding region in SARS-CoV-2 variants (Agnieszka Ruszkowska).

These analysis allow for better discussion about structural motifs conservation and possible functions.

Covariation analysis was described in section 2.5 and 4.8. Also Supporting Information 3 was added. Additional section 2.5 and 4.8 are presented below

“2.5. Covariation analysis of the sgRNA M secondary structure

Covariation analysis of the sgRNA M model presented here, utilizing the cm-builder pipeline with an alignment against 25571 Coronaviridae sequences, yielded no base pairs with statistically significant covariation. Covariation (i.e., sites of mutated sequence which maintain base pairing and 2D structure) is often used to support the potential for a RNA motif to be functional as the structure is being pre-served even when the primary sequence is not. Importantly though, a lack of covariation does not indicate a lack of potential functionality.

An analysis of the conservation of the sgRNA M model base pairs against an alignment of 453 homologous sequences showed an average base pair conservation of 97.57%, with stem SL1 (Figure 1) having an average conservation of 50.97%, stem SL2 (Figure 1) averaging 83.48%, and the remaining base pairs averaging roughly 100% conservation (Supporting_Information_3).

The high degree of conservation of most base pairs in the model may be partially responsible for the lack of detectable covariation. This lack of significant covariation is in line with previous studies [20], [15], [45], despite extensive evidence of stable, ordered secondary structure, few motifs were supported by significant covariation. Additionally, as most of the structures presented here exist within the sgRNA M coding sequence, there is additional evolutionary pressures to conserve these sequences as to not disrupt protein amino acid sequence.”

“4.8. Covariation analysis

The sequence of the in vitro probed sgRNA M underwent covariation analysis via the cm-builder pipeline and details of this process are available [20]. Briefly, cm-builder utilizes the programs INFERNAL (here, release 1.1.2 ) [51], [52]and R-scape (here, version 1.5.16) [53], [54] to make alignments of a reference sequence to homologous sequences and then cross evaluate a structural model for statistically significant covariation that maintains base pairing. The sgRNA M sequences were aligned to a previously generated fasta file [15] of 25571 Coronaviridae sequences, obtained from the ViPR database [55], [56].

Additionally, the sgRNA M sequence was queried against the nucleotide BLAST database and yielded 453 homologous sequences. These sequences were subsequently MAFFT [57] aligned with the sgRNA M sequence. From here, alignments were used to calculate conservation of nucleotides in each base pair of the model structure.”

The additional section 2.6 is presented below. A Table was also added (Table 1)

“2.6. Possible influence of nucleotide mutation on sgRNA M structure of SARS-CoV-2 variants

The SARS-CoV-2 genome constantly evolves, new mutations appear and virus variants are monitored. Changing of RNA sequence must occur in frame to not be lethal for the virus and to preserve function of proteins and also RNA structure. Emerging mutations should retain base pairs in RNA structure motifs that are important in the viral cycle. Therefore, we analyzed nucleotide mutations in the sequence of the M gene of SARS-CoV-2 variants. Table 1 displays mutations based on GISAID variants and their representatives .When compared to our model most of the nucleotide mutations occur in single stranded regions of sgRNA M structure and have no influence on the base pairing structure. Interestingly, three consistent mutations (preserving base pairs) are in helix region of motif 94-130/211-336. Consistent mutations are additional support for existing of this long-distance interactions. While two hairpin motifs in regions 153-167 and 299-329 (Figure 1) appear to have no functional significance, be-cause non-consistent mutation in these structures exclude their conservation (Table 1).”

  1. Several references need to be updated to reflect that they are no longer bioRXivs but have been published.

We thank the Reviewer for advising us to review the reference section. We checked all references of manuscript and updated them if needed. Also, all small mistakes (as absence of pages) in reference list were corrected.

I hope you find this corrections, changes and explanations satisfying.

Best regards,

Elzbieta Kierzek

Reviewer 2 Report

The manuscript has an excellent presentation of an interesting and quite unexplored domain of viral genomics. All sections are clearly described except the conclusion part that should be elaborated and generic sentences should be updated to direct the researcher in this domain to design further research. The conclusion part needs to be included in future remarks how these new findings can be relevant to the design of new therapeutics in the current stage of knowledge.

Author Response

Dear Reviewer #2,

            I am sending the revised manuscript entitled "Secondary structure of subgenomic RNA M of SARS-CoV-2” (manuscript ID: viruses-1525823). All authors of manuscript find the comments of Reviewers very helpful and interesting.

            In manuscript revision greatly contributed three additional persons, who are now included as co-authors of manuscript: Agnieszka Ruszkowska, Walter N. Moss and Collin A. O’Leary.

Below you will find our comments to suggestions:

Reviewer #2

The manuscript has an excellent presentation of an interesting and quite unexplored domain of viral genomics. All sections are clearly described except the conclusion part that should be elaborated and generic sentences should be updated to direct the researcher in this domain to design further research. The conclusion part needs to be included in future remarks how these new findings can be relevant to the design of new therapeutics in the current stage of knowledge.

We thank the Reviewer for this kind comments. The manuscript was greatly revised and new comparisons to available data and additional bioinformatics analysis suggested by Reviewers #1 and #3 were added in sections “Covariation analysis of the sgRNA M secondary structure” and “Possible influence of nucleotide mutation on ssgRNA M structure of SARS-CoV-2 variants”. There is an additional Table 1 and Supplementary_Information_3. The additional data implemented in manuscript allow for revised and better Conclusions section. In Conclusions we also added more about how these new findings can be relevant to the design of new therapeutics.

“3. Conclusions

For the first time, the secondary structure of sgRNA M was determined based on the experimental data from several chemical mapping methods and bioinformatic analysis. The secondary structure model contains unique features likely to be important for sgRNA M functions. The structure also includes several of the same distinct motifs as the genomic M fragment in the SARS-CoV-2 genome (Figure 3). Previously published reactivity of sgRNA M from structure probing in living cells supported existence of some presented structural motifs of our sgRNA M model [24]. Although covariation analysis shows no base pairs with statistically significant covariation, the mutations of gene M within SARS-CoV-2 variants is largely in agreement with the presented structure and supports long-distance helixes. This new knowledge about sgRNA M provides insights to better understand virus biology and could be used for anti-SARS-CoV-2 strategies and designing of new therapeutics. The revealed unique or same as in gRNA M structural motifs could be promising targets for antisense oligonucleotides, siRNAs and small molecules.”

The manuscript was also read by native English speakers (co-authors of manuscript) and we believe it is fluent in English and clear for readers in all parts.

I hope you find this corrections, changes and explanations satisfying.

Best regards,

Elzbieta Kierzek

Reviewer 3 Report

Soszynska-Jozwiak, Kierzek, and Kierzek use free energy minimization and experimental data from chemical mapping with DMS, CMCT, and SHAPE to report the secondary structure of subgenomic RNA M of SARS-CoV-2.  This secondary structure may be of interest to those attempting to target SARS-CoV-2.  My main concerns and questions are as follows:

  1. I wonder how impactful this will be on the field. The secondary structure of RNA of SARS-CoV-2 has been extensively studied.  How does the secondary structure of sgRNA M advance the field?  Has other sgRNA been used as a therapeutic target?
  2. This manuscript needs extensive English language editing. I don’t note all of the needed changes, but some of the needed changes are:
    • Line 24, “especially among older” needs a noun.
    • Line 32, “represent” should be “represents” (although I don’t think “represents” is the best word choice here.
    • Line 32, “possess” should be “possesses.”
    • Line 44, “is necessary” should be “are necessary.”
    • Line 61, “stem-loops structures” should be “stem-loop structures.”
    • Line 67, “sgRNAs were measured” should be “sgRNAs was measured.”
    • Line 73, the sentence “A full set…” doesn’t make sense.
    • Line 80, the sentence “This process allowed achieving…” doesn’t make sense.
    • Line 83, “was folding” should be “was folded.”
    • Line 116, “This three hairpins also occurs” should be “These three hairpins also occur.”
    • Line 126, “less thermodynamically stability” should be “less thermodynamic stability.”
    • Line 142, “These finding” should be “This finding.”
    • Line 156, “longer about 3” should be “longer by about 3.”
    • Line 156, “an additionally internal loop” should be “an additional internal loop.”
    • The Table 1 caption should be “reaction used to” and “underlined nucleotide residues.”
    • The Fig. S1 caption should read “slowly cooling to.”
    • The first sentence of the conclusion does not make sense.
    • Line 181, the closing parenthesis is missing.
    • Line 250, “lowaverage” should be “low average.”
    • Line 199, “stain” should be “strain.”
  3. Line 113, the authors state that “RNA motifs of sgRNA M model are thermodynamically stable (dG = -418,5 kcal/mol)…” First, should dG be ΔG°37?  Second, should -418,5 be -418.5?  Last, what motifs?  Does this mean the entire folded secondary structure?
  4. Line 149, a reference appears to be missing.
  5. Line 149, it not obvious what the authors are referring to when they say, “It is in agreement…” What is “it”?
  6. Line 210 should refer to Table 2.
  7. Line 216 should refer to Table 1.
  8. Line 227 should refer to Fig. S1.
  9. Line 246, what SI figure/table are the authors referring to?
  10. It doesn’t look like any of the references have page numbers.
  11. The yellow font in the Fig. 2 legend is too light to read.
  12. 3 was not mentioned in the main text.
  13. S2 was not specifically mentioned in the main text. Also, what does “231-107 nt” mean.  Should be it “nucleotides 107-231”?  Could the residue numbers be added to the x-axis?
  14. S3 was not specifically mentioned in the main text.
  15. There are several acronyms that need to be defined. These include: nsps, SHAPE, DMS (defined too late), CMCT (defined too late), HEPES, NMIA (defined too late), ADAR, FAM, and JOE.
  16. Line 66, “Nanopore direct-RNA sequencing, NAI reactivity” should be “nanopore direct-RNA sequencing (NAI) reactivity.”
  17. On line 69, the authors refer to the “model of sgRNA M.” I question the use of “model.”  Is “mimic” a better word choice?

Author Response

Dear Reviewer #2,

            I am sending the revised manuscript entitled "Secondary structure of subgenomic RNA M of SARS-CoV-2” (manuscript ID: viruses-1525823). All authors of manuscript find the comments of Reviewers very helpful and interesting.

            In manuscript revision greatly contributed three additional persons, who are now included as co-authors of manuscript: Agnieszka Ruszkowska, Walter N. Moss and Collin A. O’Leary.

Below you will find our comments to suggestions:

Reviewer #2

  1. I wonder how impactful this will be on the field. The secondary structure of RNA of SARS-CoV-2 has been extensively studied. How does the secondary structure of sgRNA M advance the field? Has other sgRNA been used as a therapeutic target?

SARS-CoV-2 viral cycle fully depends on RNA. Successful disruption of any sgRNA or gRNA will result with disruption of certain step of viral replication and in consequences degreasing amount of new virus particles. sgRNA as separate, much shorter than whole SARS-CoV-2 genome RNA, is probably more available target for different strategies as antisense strategies using oligonucleotides and small molecules targeting RNA. In our research of different viral RNA, influenza virus RNA, we showed that RNA secondary structure has a great impact on effectiveness of antisense oligonucleotides, siRNA and triplex forming PNA (Michalak, P. et al. Sci. Rep. 9, 3801 (2019); Piasecka, J. et al. Mol. Ther. - Nucleic Acids 19, 627–642 (2020); Lenartowicz, E. et al. Nucleic Acid Ther. 26, 277–285 (2016); Szabat, M. et al. Pathogens 9, 1–19 (2020), Kesy, J. et al. Bioconjug. Chem. 30, 931–943 (2019), Soszynska-Jozwiak, M. et al. Sci. Rep. 7, 15041 (2017)). Therefore it is important to know the structure of sgRNA and influence of 5’UTR on structure of coding region. Our research presented in manuscript show specific structural futures that may be connected with sgRNA function. Knowledge about secondary structure of SARS-CoV-2 sgRNAs is important for understanding virus biology.

Our research also show that in vitro study of model sgRNA combining with bioinformatics analysis, covariation and variants mutations study could contribute largely to determination of key structural features of the RNA. It is a relatively simple and fast way to know better RNA biology and for designing new therapeutics. sgRNAs were not use so far as a target for possible therapeutics although now it is obvious that inhibitors that are designed to target structural motifs that are the same in gRNA will target also sgRNA. Such knowledge about similarities and specificity of certain motifs of sgRNA comparing to gRNA – that we provided in the manuscript- is important and useful in the field.

  1. This manuscript needs extensive English language editing. I don’t note all of the needed changes, but some of the needed changes are:
  • Line 24, “especially among older” needs a noun.
  • Line 32, “represent” should be “represents” (although I don’t think “represents” is the best word choice here.
  • Line 32, “possess” should be “possesses.”
  • Line 44, “is necessary” should be “are necessary.”
  • Line 61, “stem-loops structures” should be “stem-loop structures.”
  • Line 67, “sgRNAs were measured” should be “sgRNAs was measured.”
  • Line 73, the sentence “A full set…” doesn’t make sense.
  • Line 80, the sentence “This process allowed achieving…” doesn’t make sense.
  • Line 83, “was folding” should be “was folded.”
  • Line 116, “This three hairpins also occurs” should be “These three hairpins also occur.”
  • Line 126, “less thermodynamically stability” should be “less thermodynamic stability.”
  • Line 142, “These finding” should be “This finding.”
  • Line 156, “longer about 3” should be “longer by about 3.”
  • Line 156, “an additionally internal loop” should be “an additional internal loop.”
  • The Table 1 caption should be “reaction used to” and “underlined nucleotide residues.”
  • The Fig. S1 caption should read “slowly cooling to.”
  • The first sentence of the conclusion does not make sense.
  • Line 181, the closing parenthesis is missing.
  • Line 250, “lowaverage” should be “low average.”
  • Line 199, “stain” should be “strain.”

We thank the Reviewer for careful reading of manuscript and useful comments. We corrected all of the above mistakes. Also, we revised manuscript and make all efforts to eliminate any grammar and punctuations errors and make sentences clearly understandable in whole manuscript. The text was revised to smooth English and to be well understood by readers. We would like to emphasis that two authors of manuscript who greatly contributed in revision of the manuscript are native English speakers. We also used professional English grammar checking program. We believe that revised manuscript is clearly written in fluent English.

  1. Line 113, the authors state that “RNA motifs of sgRNA M model are thermodynamically stable (dG = -418,5 kcal/mol)…” First, should dG be ΔG°37? Second, should -418,5 be -418.5? Last, what motifs?  Does this mean the entire folded secondary structure?

We thank the Reviewer for finding this mistake. This was corrected and the fragment now is:

“Our model of sgRNA M is highly structured with plenty of accessible bulges and loops (Figure 1.). RNA motifs in the sgRNA M model are thermodynamically stable and have high calculated base pair probability. ΔG°37 of the entire folded secondary structure is -416 kcal/mol.”

  1. Line 149, a reference appears to be missing.

We thank the Reviewer for finding this mistake. A reference was added.

  1. Line 149, it not obvious what the authors are referring to when they say, “It is in agreement…” What is “it”?

We thank the Reviewer for finding this unclearness. We changed this sentence to:

“This difference is in agreement with previous study about the in vivo RNA-RNA interactome of the full-length SARS-CoV-2 genome and several sgRNAs, here some structural aspects of viral RNA was also discussed [32].”

  1. Line 210 should refer to Table 2.

We thank the Reviewer for pointing this out. We added in this sentence a reference to Table, now it is a Table 3.

  1. Line 216 should refer to Table 1.

We thank the Reviewer for pointing this out. The table (now Table 2) is now referred.

  1. Line 227 should refer to Fig. S1.

We thank the Reviewer for this advice. The sentence is now referred to Figure S1.

  1. Line 246, what SI figure/table are the authors referring to?

We thank the Reviewer for the advice to precisely refer the supplementary information. We refer in this sentence to Table S1 in Supporting Information 2. This information was now added in the manuscript.

  1. It doesn’t look like any of the references have page numbers.

We thank the Reviewer for finding this mistake. We updated and corrected the list of references and now they all contain page numbers.

  1. The yellow font in the Fig. 2 legend is too light to read.

We thank the Reviewer for this advice. We changed the color from yellow to pink. The new figure was now inserted in the manuscript. Also, we checked all figures and a small mistake in presented structure was corrected in Figures 1-3.

  1. 3 was not mentioned in the main text.

We thank the Reviewer for finding this mistake. Figure 3 is now mentioned in text. It is in 5th paragraph of section 2.3.

  1. S2 was not specifically mentioned in the main text. Also, what does “231-107 nt” mean. Should be it “nucleotides 107-231”? Could the residue numbers be added to the x-axis?

And

  1. S3 was not specifically mentioned in the main text.

We thank the Reviewer for suggestion to precisely refer the supplementary information. Now all tables and figures from supplementary materials are referred along with the appropriate Supporting Information file number.

  1. There are several acronyms that need to be defined. These include: nsps, SHAPE, DMS (defined too late), CMCT (defined too late), HEPES, NMIA (defined too late), ADAR, FAM, and JOE.

We thank the Reviewer for this comment. Now all acronyms are defined when they first appear in the text.

  1. Line 66, “Nanopore direct-RNA sequencing, NAI reactivity” should be “nanopore direct-RNA sequencing (NAI) reactivity.”

We thank the Reviewer for letting us know that this sentence in not clear. We changed the sentence to:

“RNA structure probing coupled with nanopore direct-RNA sequencing were used to map sgRNAs with NAI in living cells, but the structure of sgRNA M was not proposed [24].”

  1. On line 69, the authors refer to the “model of sgRNA M.” I question the use of “model.” Is “mimic” a better word choice?

We thank the Reviewer for this suggestion. We agree that it is important to name clearly the studied RNA. We generally revised text of the manuscript. We prefer word “model” because it is “neutral”, used in in vitro studies, suggests something in vitro and “simplified” for research, as our molecule. “Mimic” could have wild meaning for readers, and are used sometimes to underline application when molecule mimic a natural molecule in biological system. Therefore we stay with the world “model” when it is needed.

I hope you find this corrections, changes and explanations satisfying.

Best regards,

Elzbieta Kierzek

Round 2

Reviewer 3 Report

Soszynska-Jozwiak et al. adequately addressed my previous concerns.  The English grammar in this version of the manuscript is much improved.  My minor concerns are:

  1. On p. 2, line 80, the authors state, “Several chemical reagents combined with a bioinformatics program allowed for prediction of a high accuracy structure.” If this is the first structure proposed, how can the authors know that it is “of a high accuracy?”
  2. 2 is mentioned in the main text before Fig. 1. They should be mentioned in order.
  3. On p. 5, line 244, what SI table or figure is being referred to?
  4. On p. 6, line 258, the acronym GISAID was not defined.
  5. On p. 7, line 328, the authors mention 6-FAM. Is that correct or should it be 5-FAM or 6-JOE?  Similarly, line 331 mentions 5-JOE.
  6. There are a few acronyms in section 4.8 that don’t appear to be defined…INFERNAL, ViPR, and MAFFT.
  7. In Table 1, the footnote descriptions should be located after the table, not in the table title.
  8. I could not find Fig S1-S3 or Table S1. The only supplemental folder available was SI_3, and these were apparently in supplemental folder SI_1 and SI_2.

Author Response

Dear Reviewer #3,

            I am sending the revised manuscript entitled "Secondary structure of subgenomic RNA M of SARS-CoV-2” (manuscript ID: viruses-1525823). All authors of the manuscript find the comments of the Reviewer very helpful. We believe that the introduced changes improved the manuscript and will make it acceptable for publication in Viruses.

Below you will find our comments to suggestions.

Reviewer #3

- Soszynska-Jozwiak et al. adequately addressed my previous concerns.  The English grammar in this version of the manuscript is much improved. 

We thank the Reviewer for this kind and positive comment.

  1. On p. 2, line 80, the authors state, “Several chemical reagents combined with a bioinformatics program allowed for prediction of a high accuracy structure.” If this is the first structure proposed, how can the authors know that it is “of a high accuracy?”

We thank the Reviewer for letting us know that this sentence is not clear. We change the sentence to:

“Several chemical reagents combined with a bioinformatics program allowed for prediction of a high confidence structure.”

  1. 2 is mentioned in the main text before Fig. 1. They should be mentioned in order.

We thank the Reviewer for finding this mistake. We corrected this and now the figures and references to them are in correct order.

  1. On p. 5, line 244, what SI table or figure is being referred to?

We thank the Reviewer for this comment that made us realize that we need to change files names in Supporting_Information_3. Supporting_Information_3 contained file “sgRNA_M_bpcount.xlsx” which is now renamed to “Table S1.xlsx” for clearness. Also, in README file the name of “sgRNA_M_bpcount.xlsx” was changed to “Table S1.xlsx”. Now we referred to Table S1, Supporting_Information_3, which is easy to find. Below is a fragment that was changed in main text of the manuscript:

“An analysis of the conservation of the sgRNA M model base pairs against an alignment of 453 homologous sequences showed an average base pair conservation of 97.57%, with stem SL1 (Figure 2) having an average conservation of 50.97%, stem SL2 (Figure 2) averaging 83.48%, and the remaining base pairs averaging roughly 100% conservation (Table S1, Supporting_Information_3).”

  1. On p. 6, line 258, the acronym GISAID was not defined.

We thank the Reviewer for pointing this out. We added definition of GISAID acronym in the manuscript text.

The sentence is now changed to:

“Table 1 displays mutations based on SARS-CoV-2 variants and their representatives deposited in Global Initiative on Sharing Avian Influenza Data (GISAID).”

  1. On p. 7, line 328, the authors mention 6-FAM. Is that correct or should it be 5-FAM or 6-JOE? Similarly, line 331 mentions 5-JOE.

We thank the Reviewer for this comment and question. We used 6-FAM and 5-JOE. The question made us aware of typo mistake that is now corrected in the manuscript text and in Table 3 title.

  1. There are a few acronyms in section 4.8 that don’t appear to be defined…INFERNAL, ViPR, and MAFFT.

We thank the Reviewer for this comment. We added definition of these acronyms in the manuscript text.

The changed paragraph is below:

“4.8. Covariation analysis

The sequence of the in vitro probed sgRNA M underwent covariation analysis via the cm-builder pipeline and details of this process are available [20]. Briefly, cm-builder utilizes the programs INFERence of RNA ALignment (INFERNAL) (here, release 1.1.2 ) [51], [52]and R-scape (here, version 1.5.16) [53], [54] to make alignments of a reference sequence to homologous sequences and then cross evaluate a structural model for statistically significant covariation that maintains base pairing. The sgRNA M sequences were aligned to a previously generated fasta file [15] of 25571 Coronaviridae sequences, obtained from the Virus Pathogen Database and Analysis Resource (ViPR database) [55], [56].

Additionally, the sgRNA M sequence was queried against the nucleotide BLAST (Basic Local Alignment Search Tool) database and yielded 453 homologous sequences. These sequences were subsequently MAFFT (Multiple Alignment using Fast Fourier Transform) [57] aligned with the sgRNA M sequence. From here, alignments were used to calculate conservation of nucleotides in each base pair of the model structure.”

  1. In Table 1, the footnote descriptions should be located after the table, not in the table title.

We thank the Reviewer for this comment. We corrected table and the footnote description in Table 1 according to the remark.

  1. I could not find Fig S1-S3 or Table S1. The only supplemental folder available was SI_3, and these were apparently in supplemental folder SI_1 and SI_2.

We thank the Reviewer for carefully reading and comments. The Supplementary files: Supporting_Information_1, Supporting_Information_2 and Supporting_Information_3 were uploaded and we believed they should be available. We are sorry if they were not available from some reason, maybe technical.

Right now we carefully checked all Supplementary files submitted. We also checked the reference to Supplementary file and their figures and tables. Now it should be easy for readers to find appropriate file and appropriate figure or table.

I hope you find these corrections and changes satisfying.

Best regards,

Elzbieta Kierzek